# Controllable Connection of Fe_2_Se_3_ Double Chains and Fe(dien)_2_ Complexes for Organic–Inorganic Hybrid Ferrimagnet with a Large Coercivity

**DOI:** 10.3390/nano13030487

**Published:** 2023-01-25

**Authors:** Xiaolei Shang, Xiaoling Men, Qifeng Kuang, Shaojie Li, Da Li, Zhidong Zhang

**Affiliations:** 1Shenyang National Laboratory for Materials Science, Institute of Metal Research, Chinese Academy of Sciences, 72 Wenhua Road, Shenyang 110016, China; 2School of Materials Science and Engineering, University of Science and Technology of China, 72 Wenhua Road, Shenyang 110016, China; 3Instrumental Analysis and Research Center, Dalian University of Technology, Panjin 124221, China

**Keywords:** organic–inorganic hybrid materials, chemical solution method, Fe_2_Se_3_ double chains, coercivity

## Abstract

Organic–inorganic hybrid materials built by inorganic and organic building units have attracted intensive interest in the past decades due to unique chemical and physical properties. However, rare organic–inorganic hybrid materials show excellent permanent magnetic properties. Here, we develop a facile chemical solution method to bottom-up synthesize a new hybrid (Fe_2_Se_3_)_2_[Fe(dien)_2_]_0.9_. This hybrid phase with the space group *P2_1_/c* (14) possesses a rodlike shape with a diameter of 100–2000 nm and a length of 5–50 µm. The hybrid rods are ferrimagnetic with a Curie temperature (*T*_C_) of 11 K. They show a high coercivity (*H*_C_) of 4.67 kOe and a saturation magnetization (*M*_S_) of 13.5 emu/g at 2 K. Compared with orthorhombic (FeSe_2_)_2_Fe(dien)_2_, the excellent magnetic performance of the hybrid rods is ascribed to the monoclinic hybrid structure built by Fe(dien)_2_ complexes and Fe_2_Se_3_ double chains. Our study provides guidance for connecting inorganic fragments of FeSe_2_ single chains, Fe_2_Se_3_ double chains or β-Fe_3_Se_4_ layers with Fe(dien)_2_ complexes for organic–inorganic hybrid phases with varied crystal structures and magnetic properties.

## 1. Introduction

Over the past years, the development of organic magnetic materials has been intensively explored for both fundamental research and technological applications [1]. Organic–inorganic hybrid materials combining both organic and inorganic components exhibit diverse structures and unique physical and chemical properties. The advantages of organic–inorganic hybrid materials have attracted strong interest in potential applications in electronic and optical devices [2,3], protective coatings [4], catalysts [5], high-temperature superconductors [6], and so on. Many organic–inorganic hybrid materials are antiferromagnetic or weak ferromagnetic (ferrimagnetic) with low magnetic ordering temperatures [7,8,9,10,11]. Separation of magnetic inorganic fragments, such as one-dimensional (1D) chains [12,13] and 2D plates [14], by organic molecules or complexes results in the magnetism of organic–inorganic hybrid materials, and their magnetic ordering temperatures are usually much below room temperature [15,16,17,18]. Nevertheless, there are rare magnetic organic–inorganic hybrid materials, for example, (Fe_2_Se_3_)_4_[Fe(tepa)] (tepa = tetraethylenepentamine) [19], (β-Fe_3_Se_4_)_4_[Fe(teta)_1.5_] (teta = triethylenetetramine) [20] and (NH_3_–CH_2_–C_6_H_4_CO_2_H)[SnCl_6_] [21], showing their magnetic ordering temperatures higher than room temperature. Moreover, Fe^3+^ vacancy doping gives rise to a room-temperature long-range ferrimagnetic (FIM) order in an organic–inorganic hybrid (FeSe_2_)_2_Fe(dien)_2_ (dien = diethylenetriamine) [22]. Noncompensated spins in the (Fe_0.86_Se_2_)_2_Fe(dien)_2_ due to Fe^3+^ vacancies are produced by controllable self-assemble of 1D FeSe_2_ short chains and Fe(dien)_2_ complexes [22], which is different from the “chemical scissors” model to bond isolated single FeSe_2_ chains with Fe(dien)_2_ complexes [13,17]. Moreover, most organic–inorganic hybrid materials have low coercivities. It is very difficult to improve coercivity for organic and inorganic hybrid materials. It is preferable to control the stoichiometry of the inorganic component and the combination of inorganic and organic fragments for excellent magnetic performance. From the point of view of magnetism, it is necessary to develop a “bottom-up self-assemble” that is able to enhance the magnetic properties of organic–inorganic hybrid materials through tuning magnetic interactions between Fe-amine complexes and Fe*_x_*Se building blocks in different dimensions [23]. In comparison with a “top-down” method to synthesize organic–inorganic hybrid materials [24], the “bottom-up self-assemble” is more conducive to manipulating the assemble process at the atomic/molecular level. Recently, quasi-1D spin-ladder compounds, *M*Fe_2_Se_3_ (*M* = Ba, Cs), have attracted considerable interest [25,26,27,28], in which iron and selenium form Fe_2_Se_3_ double chains. FeSe_4_ tetrahedra sharing edges in Fe_2_Se_3_ double chains are much different from the FeSe_2_ single chains mentioned above. More recently, we reported a new type of iron chalcogenide-based superconducting system, which originates from a suppression of the long-range FIM order of a parent phase, [Fe(tepa)](*β*-Fe_2_Se_3_)_4_, through simultaneously tuning the host and the spacer layers [29].

In this work, we report a facile solution-based synthetic route to fabricate a new organic–inorganic hybrid magnet built by organic Fe(dien)_2_ and inorganic Fe_2_Se_3_ double chains. The (Fe_2_Se_3_)_2_[Fe(dien)_2_]_0.9_ hybrid forms with a rodlike shape, which is a single phase in a monoclinic crystal structure with the space group *P2_1_/c* (14). Temperature dependence of magnetization reveals the ferrimagnetism of the (Fe_2_Se_3_)_2_[Fe(dien)_2_]_0.9_ rods with the Curie temperature (*T*_C_) of 11 K. The coercivity (*H*_C_) and saturation magnetization (*M*_S_) of the (Fe_2_Se_3_)_2_[Fe(dien)_2_]_0.9_ magnet is 4.67 kOe and 13.5 emu/g, respectively, at 2 K. Our study not only provides guidance for developing a chemical solution method for controllable fabrication of a new type of organic–inorganic hybrid materials but also extends a potential application as an organic permanent magnet due to high coercivity of this organic–inorganic hybrid magnet.

## 2. Materials and Methods

### 2.1. Chemicals

Iron(III) acetylacetonate [Fe(acac)_3_, 98%] was purchased from Aladdin reagent company (Shanghai, China). Diethylenetriamine (dien, 99%), acetone (99.7%), and isopropyl alcohol (99.7%) were purchased from Sinopharm Chemical Reagent Co., Ltd. (Shenyang, China). Se powder (99.99%) was purchased from Macklin Biochemical Co., Ltd. (Shanghai, China). All chemicals were used without further purification.

### 2.2. Synthesis of Monoclinic (Fe_2_Se_3_)_2_[Fe(dien)_2_]_0.9_ Nanorods

In a typical synthesis of (Fe_2_Se_3_)_2_[Fe(dien)_2_] hybrid nanorods, 3.25 mmol Fe(acac)_3_, 4 mmol Se powder and 40 mL dien were mixed into a 100 mL quartz crucible in a stainless- steel autoclave under magnetic stirring throughout the entire reaction process. Under an argon gas flow, the mixture was kept at room temperature for 1 h, and then was heated to 393 K and kept at this temperature for 1 h to remove moisture and oxygen. Then the temperature was raised to 433 K and kept for 3 h in order that all raw materials were dissolved into the solution. The solution was subsequently heated to 503 K at a rate of 0.4 K·min^−1^ and maintained for 3 days. At room temperature, the product was centrifuged at 8000 rpm for 5 min. A ThermoFisher inductively coupled plasma (ICP) spectroscopy gave a concentration of 0.0072 and 0.06 mg mL^−1^, respectively, for Fe and Se ions left in the light-yellow transparent supernatant above the precipitate, suggesting that all the Fe and Se atoms in the starting materials were combined in the hybrid product. The precipitate was washed by 20 mL acetone and 5 mL isopropyl alcohol the first time and then rewashed in 20 mL acetone four times. The product was dried in a vacuum for further characterization.

For comparison, Fe_3_Se_4_(dien)_2_ particles and (Fe_3.5_Se_4_)-dien plates were synthesized in a manner similar to the synthesis of monoclinic (Fe_2_Se_3_)_2_[Fe(dien)_2_]_0.9_ rods except that the reaction temperature was set to 473 K and 513 K, respectively. The Fe:Se stoichiometry for different hybrid material was determined by the amounts of the Fe and Se precursors. However, a formula of the hybrid plates cannot be determined due to the hybrid phase being mixed with some decomposed tetragonal Fe_x_Se.

### 2.3. Characterization

Powder X-ray diffraction (XRD) was performed on a Rigaku D/Max-2400 diffractometer (Rigaku Inc., Tokyo, Japan) with a Cu K_α_ radiation source (λ = 0.154056 nm). The software Fullprof was used to refine the structure of the hybrid product. The size, morphology and microstructure of the as-synthesized products were observed by a JSM 6301F field-emission scanning electronic microscope (SEM) (JEOL Inc., Japan) and a Tecnai G2 F20 transmission electronic microscope (TEM) (FEI Inc., Hillsboro, OR, USA) at 200 kV. The elemental compositions of products were determined by Oxford X-ray energy dispersive (EDX) spectroscopy and inductively coupling plasma spectrometry (ICP). Fourier transform infrared (FTIR) spectrum was recorded on a Nicolet iN10 MX & iS10 spectrometer using the KBr pellet technique (Thermo Fisher Inc., Waltham, MA, USA). Thermal gravimetric analysis (TGA) was performed on an STA6000 thermal analyzer (PerkinElmer Inc., Waltham, MA, USA) under N_2_ flow with a heating rate of 10 K·min^−1^ between 300 and 800 K. Fe *L*-edge X-ray absorption spectroscopy (XAS) measurements were performed at Beamline 08U1A of the Shanghai Synchrotron Radiation Facility (SSRF) in total electron yield (TEY) mode at room temperature. Magnetic hysteresis loops and temperature-dependent magnetization in the field-cooling (FC) mode (at *H* = 100 Oe) were carried out using a superconducting quantum interference device (SQUID) (Quantum Design Inc., San Diego, CA, USA).

## 3. Results

### 3.1. Structural Properties

SEM (Figure 1a,b) and TEM (Figure 1d) images illustrate that the hybrid product synthesized at 503 K for 3 days possesses a rodlike shape with a size distribution of about 100–2000 nm for diameter and 5–50 µm for length. The microstructure at the edge of a rod with a large diameter reveals cleavage grooves (Figure 1b), suggesting that the hybrid nanorods with smaller diameters are cleaved down from the rods with large diameters. The selected area electron diffraction (SAED) pattern of a hybrid nanorod in the inset of Figure 1d reveals periodic crystal structure along the longitudinal direction. Compared with the previous iron vacancy doped hybrids [22], clear diffraction points in the SAED pattern show a single crystal feature of the hybrid nanorod, possibly due to the absence of iron vacancies in the hybrid rods. This was supported by the composition analyses. The EDX spectrum as shown in the inset of Figure 1c indicates the presence of Fe, Se, C and N elements, and Fe and Se are the only heavy elements in the hybrid rods. Elemental mapping of a single nanorod (Figure 1e) shows the even distribution of Fe, Se, C, and N in the hybrid rod (Figure 1f–i). The relative molar ratio of Fe to Se in the hybrid rods was determined to be 44.84/55.16 (3.25/4) by EDX, very similar to the value of 3.27/4 determined by the ICP technique. The FTIR spectrum (Figure 2) presents characteristic peaks of dien featured by the vibration bands of –CH_2_–, –NH_2_, and –NH. It reveals the incorporation of organic dien molecules in the hybrid structure. Similar to the previous iron vacancy doped hybrids [22], the bands of –NH_2_ shift from 3360 cm^−1^ for the asymmetric stretching vibration of free dien molecules to 3289 cm^−1^ for that of the hybrid rods and from 3280 cm^−1^ for the symmetric stretching vibration of free dien molecules to 3217 cm^−1^ for that of the hybrid rods, which correspond to the chemical coordination of –NH_2_ groups with Fe^2+^ ions in the present hybrid rods.

TGA and dWeight/d*T* curves in Figure 3 show a weight loss (~6.7 wt%) from room temperature to the onset decomposition temperature (*T*_onset_) ~540 K due to a small amount of free dien and other organic solvents absorbed on the surface of the hybrid rods. The *T*_onset_ value is obviously higher than that of the previously reported (Fe_0.86_Se_2_)_2_Fe(dien)_2_ [22]. Between 540 K and ~585 K, a sharp weight loss of about 15.9 wt% should be ascribed to releasing the bonded dien from the hybrid rods, while a slow weight loss (5.3 wt%) between 585 K and the end temperature (*T*_end_) of ~700 K is due to the loss of the left dien and a small amount of Se. To obtain a precise content of dien in the hybrid rods, we performed the experiments as follows: At first, clean Al_2_O_3_ crucibles were calcined to 1100 K for 24 h in an air atmosphere. The Al_2_O_3_ crucibles were used to contain the hybrid rods for the subsequent thermal decomposition. The masses of an Al_2_O_3_ crucible and the hybrid rods were weighed together at different steps. The mass of the Al_2_O_3_ crucible was deducted for convenience. The hybrid rods with an initial mass of 68.6 mg were slowly heated to 580 K in an Ar flow for 24 h, at which temperature the hybrid rods were completely decomposed through releasing coordinated dien but without losing Se atoms from the decomposed product. The mass of the decomposition product was 51.1 mg. Therefore, the weight loss was determined to be about 25.4 wt%, which is smaller than the total weight loss obtained by TGA without loss of Se. If we subtract the loss of free dien and other organic solvents absorbed on the surface of the hybrid rods, the loss of coordinated dien should be about 18.7 wt%. Taking into account that there are 18.7 g of dien and 74.6 g of inorganic Fe_3.25_Se_4_ in 100 g of hybrid rods, the molar ratio *y* between dien to Fe_3.25_Se_4_ in the hybrid rods could be calculated by the equation:$$y=\frac{18.7/M_{dien}}{74.6/M_{\mathrm{Fe}3.25\mathrm{Se}4}},$$
where $M_{\mathrm{dien}}$ and $M_{\mathrm{Fe}3.25\mathrm{Se}4}$ are the molecular weights of the organic component (dien) and the inorganic component (Fe_3.25_Se_4_), respectively. The *y* for the hybrid rods is ~1.2 and the formula of the hybrid rods is determined as (Fe_3.25_Se_4_)(dien)_1.2_.

Powder XRD pattern of the hybrid rods (Figure 4) reveals that all Bragg diffraction peaks can be refined and indexed by using the monoclinic structure with the space group *P2_1_/c* (14). The corresponding refinement process gave good *R* factors (*R*_p_ = 8.52%, *R*_wp_ = 7.45% and *χ*^2^ = 2.72), and refined room-temperature lattice parameters of the hybrid material are *a* = 11.397(7) Å, *b* = 19.376(3) Å, *c* = 11.185(6) Å and *β* = 105.04°. The crystal structure of the hybrid rods is the same as that for Fe_3_Se_4_(tren) (tren = tris(2-aminoethyl)amine) [13] but different from those for previous orthorhombic Fe_3_Se_4_(dien)_2_ [13] and iron vacancy doped (Fe_0.86_Se_2_)_2_Fe(dien)_2_ [22], in which two independent subsystems have been identified: 1D FeSe_2_ chains and Fe(dien)_2_ complex. Because two dien molecules with six nitrogen atoms have bonded well with a Fe^2+^ to form a [Fe(dien)]^2+^ complex, the Fe(dien)_2_ complex should be the only organic fragment for the orthorhombic Fe_3_Se_4_(dien)_2_ [13] and the monoclinic hybrid rods. Therefore, a change of crystal structure from the orthorhombic Fe_3_Se_4_(dien)_2_ [13] to the present monoclinic hybrid rods should be ascribed to a bit increase of Fe atoms in the inorganic fragment of hybrid rods. The Fe_2_Se_3_ superstructure is the closest fragment to FeSe_2_ in chemical composition [23]. Therefore, a formula of the hybrid rods may be written as (Fe_1.99_Se_3_)_2_[Fe(dien)_2_]_0.9_. Taking into account two independent subsystems consisting of the (Fe_2_Se_3_)^2−^ double chains and the Fe^2+^(dien)_2_ complexes, there are both Fe^3+^ and Fe^2+^ ions in the hybrid rods similar to the case of the orthorhombic Fe_3_Se_4_(dien)_2_ [13]. Figure 5 represents the Fe *L*-edge X-ray absorption spectrum (XAS) line of the monoclinic hybrid rods in total electron yield (TEY) mode, along with that of commercial Fe_3_O_4_ as a reference for Fe^3+^ and Fe^2+^ oxidation states. As a guide to the valence states, the line shapes of the sample are compared with that of reference, indicating the signatures of Fe^3+^ and Fe^2+^. This reveals that there are both trivalent and bivalent Fe ions in the as-synthesized monoclinic hybrid rods. It should be noted that information on the monoclinic hybrid phase was seldom mentioned in the previous transition-metal chalcogenides-dien systems [13,17,20,22,30,31,32,33].

### 3.2. Growth Mechanism

Similar to the Fe vacancies doped orthorhombic hybrid cuboids [22], the monoclinic hybrid rods have been synthesized by using soluble iron and selenium precursors and organic dien. Although inorganic FeSe*_x_* building blocks, such as 1D FeSe_2_ chains and 2D β-Fe_3_Se_4_ layers, play an important role in the formation of the FeSe*_x_*-amine hybrid materials [13,17,20,22], there is no precise information on how the 1D FeSe_2_ chains change to the 2D β-Fe_3_Se_4_ layers. In the solution syntheses, the dien molecules should play three important roles in the solvent, the reductive agent and the precursor at the same time [22]. Se atoms and a part of Fe^3+^ ions can be reduced by dien to Se^2−^ ions and Fe^2+^ ions, respectively. Meanwhile, the redox reactions produce nitrogen, serving as the production of the oxidation [21,34]. Based on a bottom-up growth mechanism, Figure 6 illustrates three combination modes of organic Fe(dien)_2_ complexes with (I) 1D FeSe_2_ chains, (II) Fe_2_Se_3_ double chains and (III) 2D β-Fe_3_Se_4_ layers, respectively, in the present reaction system, following the reaction formulas:Fe^2+^ + 2dien = Fe^2+^(dien)_2_(1)
Fe^3+^ + 2Se^2−^ = [FeSe_2_]^−^(2)
2Fe^3+^ + 4Se^2−^ + Fe^2+^ + 2dien = [FeSe_2_]_2_[Fe(dien)_2_](3)
3[FeSe_2_]_2_[Fe(dien)_2_] + 0.75Fe^2+^ = 2[Fe_2_Se_3_]_2_[Fe(dien)_2_]_0.875_ + 2.5dien(4)

The Fe^2+^(dien)_2_ complexes are formed by coordinating the Fe^2+^ ions with dien molecules, while the FeSe_2_ chains are initially created by the reaction of Se^2−^ and Fe^3+^ ions, following the reaction formulas (1) and (2). Combinations of FeSe_2_ chains and Fe^2+^(dien)_2_ complexes are driven by the Se∙∙∙H–N interactions due to strong electronegativity of Se atoms [22]. Therefore, the synthesis of orthorhombic [FeSe_2_]_2_[Fe(dien)_2_] with a cuboid shape can be proposed as the growth model (I) through a self-assembly reaction of 1D FeSe_2_ chains and Fe^2+^(dien)_2_ complexes following the reaction formula (3). The powder XRD pattern of the hybrid cuboids (Figure 7a) indicates that all Bragg diffraction peaks can be refined and indexed by using the orthorhombic structure with the space group *C2221* (No. 20). Such an XRD pattern reveals the same crystal structure as that of Fe_3_Se_4_(dien)_2_ [13]. The corresponding refinement process gave good *R* factors (*R_p_* = 18.3%, *R_wp_* = 18.7% and *χ^2^* = 6.48) for the hybrid cuboids. Refined lattice parameters at room temperature of the hybrid cuboids are 9.226(6) Å for *a* axis, 18.000(0) Å for *b* axis, and 11.610(7) Å for *c* axis, about 0.94% and 0.58% expansion along the *a* and *c* axes and 0.67% shrinkage along the *b* axis when compared with those of Fe_3_Se_4_(dien)_2_ [13]. The orthorhombic hybrid particles are in the paramagnetism in good agreement with previous Fe_3_Se_4_(dien)_2_ [13]. The magnetic properties are not shown here.

Given the redox reaction system at a higher temperature (503 K), Fe^3+^ ions are preferably reduced to Fe^2+^ ions. When the molar ratio of Fe and Se atoms in the starting materials changes from 3:4 to 3.25:4, a diffusion of more iron atoms into the orthorhombic [FeSe_2_]_2_[Fe(dien)_2_] would occur following the reaction formula (4), resulting in the monoclinic hybrid rods (Figure 4). A reasonable molecular formula for the hybrid rods is proposed as [${\mathrm{Fe}}_{1.125}^{3+}{\mathrm{Fe}}_{0.875}^{2+}{\mathrm{Se}}_{3}^{2-}$]_2_[${\mathrm{Fe}}^{2+}$(dien)_2_]_0.875_ according to electrical neutrality principles. The content of 0.875 dien/cell is obtained by balancing chemical Equation (4), which is almost the same as the experimental value of 0.9 dien/cell for the present hybrid rods determined above. Assuming that the small difference between the calculated and the experimental values is in range of the analytical error, we propose that the monoclinic hybrid rods are built by Fe_2_Se_3_ double chains and Fe(dien)_2_ complexes in the model (II).

Moreover, further increases in the molar ratio of Fe and Se to 3.5:4 and the reaction temperature to 513 K result in the hybrid nanoplates. Figure 8 shows the morphology transformation from particles for the orthorhombic phase to rods for the monoclinic phase and plates for the (Fe_3.5_Se_4_)-dien. Figure 7b presents the XRD pattern of (Fe_3.5_Se_4_)-dien hybrid plates. Two strong peaks at low 2*θ* angles of 8.48° and 17.00° with lattice spacings of 10.42 and 5.21 Å reveal the presence of period inorganic layers separated by the Fe(dien)_2_ complex, similar to previous hybrid plates with a tetragonal β-Fe_3_Se_4_ superstructure [20] and an inorganic Fe_0.9_Co_0.1_S_1.2_ unit [31]. Because the hybrid plates are mixed by an impurity of Fe_x_Se with an uncertain quality, the formula for the hybrid phase is not determined by the same method for the hybrid rods. However, an ordered tetragonal superstructure is determined in the presence of the hybrid plates by a SAED image (the inset of Figure 8d), which is similar to that of the inorganic Fe_0.9_Co_0.1_S_1.2_ unit [31] and the tetragonal β-Fe_3_Se_4_ superstructure [20]. Taking into account the composition, this ordered superstructure is suggested as the tetragonal β-Fe_3_Se_4_. Therefore, the growth model (III) can be illustrated by the combination of the inorganic tetragonal β-Fe_3_Se_4_ superstructures and the Fe(dien)_2_ complexes for the synthesis of hybrid nanoplates in the iron selenide-dien hybrid system. Efforts to examine the crystal structure of (Fe_3.5_Se_4_)-dien hybrid plates are currently in progress.

As shown in Figure 6, varied inorganic building units are key factors for syntheses of corresponding hybrid products, which depend on the Fe/Se stoichiometry and the reacting temperatures. In the solution reaction system, iron-diffusion reactions, such as the reaction formulas (3) and (4), are driven by different reaction temperatures. Figure 4 reveals a single phase of hybrid rods with a monoclinic structure, which is built by Fe_2_Se_3_ double chains and Fe(dien)_2_ complexes at a molecular level and in a controllable manner. An improper reaction condition will result in mixtures of several hybrid materials but not a single phase.

### 3.3. Magnetic Properties

Figure 9 represents the magnetic properties of the monoclinic hybrid rods. In contrast to the stoichiometric [FeSe_2_]_2_[Fe(dien)_2_] that is paramagnetic [13], the temperature dependence of the magnetization in the FC process for the monoclinic hybrid rods shows a magnetic transition at the Curie temperature (*T*_C_) of 11 K (inset of Figure 9). The *T*_C_ value is determined by the point of intersection of the two tangents around the inflection point of the FC magnetization curve. Such a *T*_C_ value is much lower than that of the iron vacancy doped (Fe_0.86_Se_2_)_2_Fe(dien)_2_ [22], revealing that the ferrimagnetism is independent on the uncompensated magnetic iron ions. There is no magnetic transition above 11 K, suggesting that the origin of ferrimagnetic behavior of this hybrid material could not be caused by traces of ferromagnetic impurity Fe_3_O_4_ with the *T*_C_ of about 860 K and ferrimagnetic iron selenides, such as Fe_3_Se_4_ and Fe_7_Se_8_. It is well known that the *T*_C_ values for Fe_3_Se_4_ and Fe_7_Se_8_ are about 320 K [35] and 450 K [36], respectively. The magnetic hysteresis loops of the hybrid rods in Figure 9 further exhibit the ferromagnetic feature in a temperature ranging from 2 to 10 K. The magnetization at 50 kOe (*M*50) and coercivity of the hybrid rods are 13.6 emu/g and 4.67 kOe, respectively, at 2 K. Such a *M*50 of the hybrid rods is obviously larger than the saturation magnetization value of 5.6 emu/g for ferromagnetic (CH_3_NH_3_)_2_CuCl_4_ at 10 K [7], of 6 emu/g for FIM [Fe_14_Se_16_](tepa) hybrid plates at 10 K [29] and 0.01 emu/g for (NH_3_–CH_2_–C_6_H_4_CO_2_H)[SnCl_6_] at 300 K [21]. Moreover, the high coercivity of the hybrid rods is rare in organic–inorganic hybrid materials, for example, 30 Oe for the (PEA)_2_CuCl_4_ (PEA = C_6_H_5_C_2_H_4_NH_3_) film at 5 K [9], 404 Oe for the (NH_3_–CH_2_–C_6_H_4_CO_2_H)[SnCl_6_] at 300 K [21], 1.5 kOe for TBA intercalated NiPS_3_ (TBA = tetrabutylammonium) at 5 K and 2.2 kOe for cobaltocenium ions (Co(Cp)_2_^+^ (Cp = cyclopentadienyl ring C_5_H_5_^−^) intercalated NiPS_3_ at 5 K [24]. We suggest that the large coercivity results from large uniaxial magnetocrystalline anisotropy of the hybrid rods [37]. As the temperature rises, the coercivity significantly decreases to 1.65 kOe at 5 K and about 20 Oe at 10 K. It is obvious that the magnetic field dependence of magnetization is in the ferromagnetic characteristics in a low magnetic field range, but in a linear magnetization in the high magnetic field range. The linear magnetization curves in the high field range may indicate the existence of a paramagnetic or antiferromagnetic component because of the nonsaturation feature, even at the magnetic field of 50 kOe. Such a linear magnetization process was ascribed to either weakened sublattice or intersublattice spins disoriented by thermal motion [35] or the noncollinear spin alignments in the magnetic system. The linear field dependence of the magnetization at higher fields is not contrary to the FIM compounds, in which strong antiferromagnetic coupling between different magnetic sublattices prevents their full alignment by the magnetic field [20]. The remanant magnetization (*M*_r_) is 8 emu/g at 2 K, which results in a good magnetic remanence ratio (*M*_r_/*M*50) of ~0.6. The excellent magnetic properties may be ascribed to the quasi-one-dimensional Fe_2_Se_3_ double chains in the hybrid rods, which is potential to be applied in the new type of permanent magnets.

Quasi-1D spin-ladder compounds, such as *M*Fe_2_Se_3_ (*M* = Ba, Cs) [25,26,27,28], have attracted considerable interest. Either block magnetism [25] or stripelike magnetism [27] was proposed as the magnetic origins of these ladder compounds, where the magnetic moments couple ferromagnetically (or antiferromagnetically) along the rung (or the leg) direction. Because spin ladders in copper oxides shed light on the mechanism of superconductivity, a study on an analog with ladder geometry among ferrous compounds is highly interesting. It is obvious that all the iron atoms in BaFe_2_Se_3_ are ferric, while those in CsFe_2_Se_3_ are composed of ferric and ferrous ions with a molar ratio of 1:1. In our case, the molecular formula of [${\mathrm{Fe}}_{1.125}^{3+}{\mathrm{Fe}}_{0.875}^{2+}{\mathrm{Se}}_{3}^{2+}$]_2_[${\mathrm{Fe}}^{2+}$(dien)_2_]_0.875_ reveals the iron composition, which enables creation of a magnetic structure that is different from those for the antiferromagnetic BaFe_2_Se_3_ and CsFe_2_Se_3_ due to the varied content of ferric and ferrous ions. This monoclinic hybrid material has been classified by the large magnetization and coercivity at 2 K as a ferrimagnetic compound. In comparison with the Fe_3_Se_4_(dien)_2_ with 1D ${\mathrm{Fe}}^{3+}{\mathrm{Se}}_{2}^{2+}$ single chains [13], the ferric and ferrous ions in ${\mathrm{Fe}}_{1.125}^{3+}{\mathrm{Fe}}_{0.875}^{2+}{\mathrm{Se}}_{3}^{2-}$ double chains should be responsible for the ferrimagnetic interactions. Moreover, we easily see that the ${\mathrm{Fe}}_{1.125}^{3+}{\mathrm{Fe}}_{0.875}^{2+}{\mathrm{Se}}_{3}^{2-}$ double chains in present hybrid rods are different from ${\mathrm{Fe}}_{2}^{3+}{\mathrm{Se}}_{3}^{2-}$ in BaFe_2_Se_3_ and ${\mathrm{Fe}}^{3+}{\mathrm{Fe}}^{2+}{\mathrm{Se}}_{2}^{2-}$ in CsFe_2_Se_3_. Due to a diffusing reaction in the formation of [${\mathrm{Fe}}_{1.125}^{3+}{\mathrm{Fe}}_{0.875}^{2+}{\mathrm{Se}}_{3}^{2-}$]_2_[${\mathrm{Fe}}^{2+}$(dien)_2_]_0.875_, the distributions of Fe^3+^ and Fe^2+^ in the ${\mathrm{Fe}}_{1.125}^{3+}{\mathrm{Fe}}_{0.875}^{2+}{\mathrm{Se}}_{3}^{2-}$ double chains are not clear. Efforts to examine the magnetic structure of [Fe_2_Se_3_]_2_[Fe(dien)_2_]_0.9_ hybrid rods with determined atom coordinates are currently in progress.

## 4. Conclusions

We developed a facile protocol for bottom-up synthesizing organic–inorganic hybrid materials by connecting inorganic fragments of FeSe_2_ single chains, Fe_2_Se_3_ double chains or β-Fe_x_Se layers with Fe(dien)_2_ complexes. These materials possess varied crystal structures and magnetic properties. Of note, the monoclinic (Fe_2_Se_3_)_2_[Fe(dien)_2_]_0.9_ hybrid rods with the space group *P2_1_/c* (14) are ferrimagnetic and the Curie temperature (*T*_C_) is determined to be ~11 K. The hybrid rods show a high coercivity (*H*_C_) of 4.67 kOe and a saturated magnetization (*M*_S_) of 13.5 emu/g at 2 K, which should be ascribed to the ferrimagnetic coupling between the organic Fe(dien)_2_ complexes and inorganic Fe_2_Se_3_ double chains. Large coercivity and magnetic remanence ratios make the monoclinic hybrid rods an excellent organic permanent magnet.

## Figures and Tables

**Figure 1 nanomaterials-13-00487-f001:**
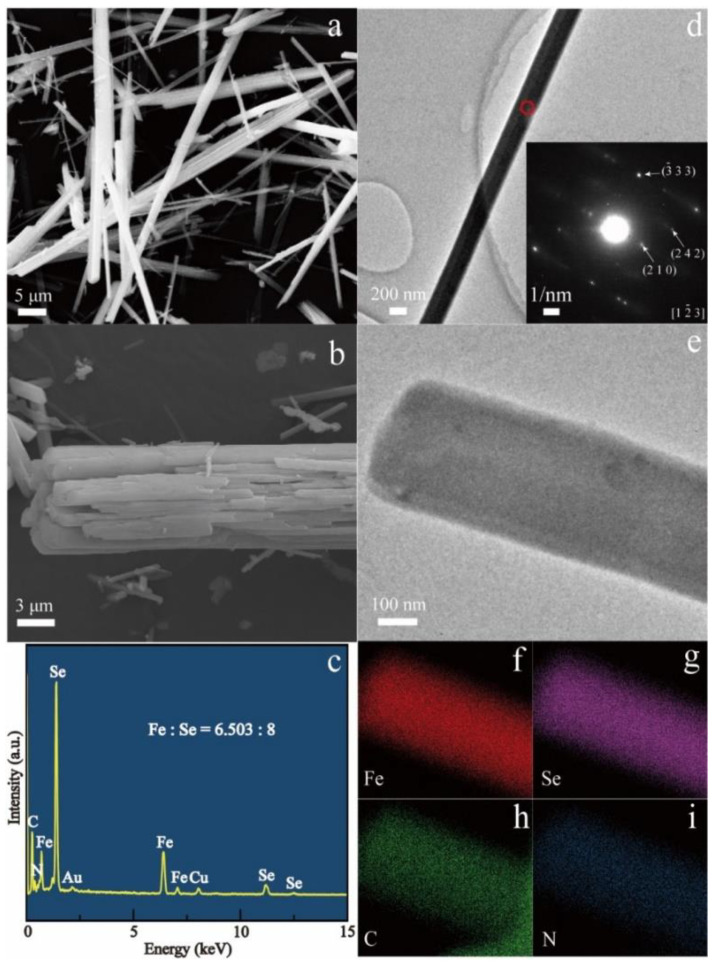
(**a**,**b**) SEM images and (**c**) the corresponding EDX spectrum of the (Fe_2_Se_3_)_2_[Fe(dien)_2_]_0.9_ hybrid rods. (**d**) Typical TEM image of a single hybrid rod. Inset shows the corresponding SAED pattern. (**e**) A STEM image and corresponding element mapping of (**f**) Fe, (**g**) Se, (**h**) C and (**i**) N for a single (Fe_2_Se_3_)_2_[Fe(dien)_2_]_0.9_ hybrid rod.

**Figure 2 nanomaterials-13-00487-f002:**
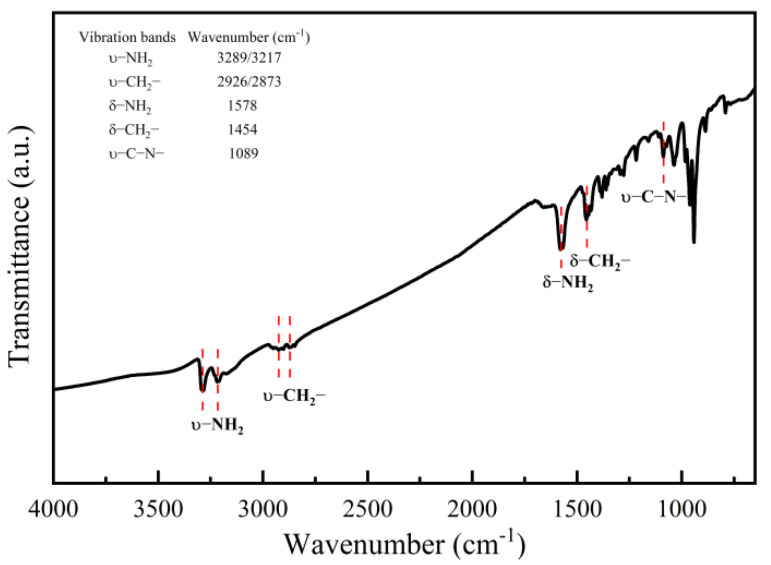
FTIR spectrum of the (Fe_2_Se_3_)_2_[Fe(dien)_2_]_0.9_ hybrid rods. The inset shows the vibration bands of –CH_2_–, –NH_2_, and –CN–.

**Figure 3 nanomaterials-13-00487-f003:**
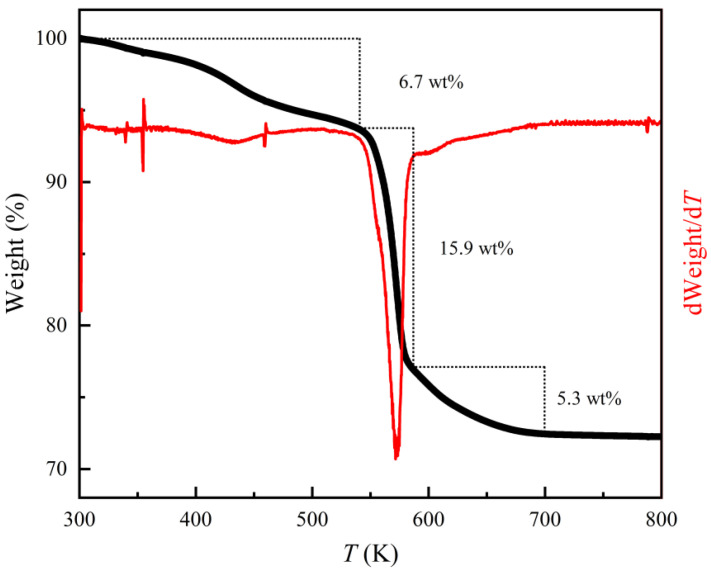
TGA and dWeight/d*T* curves of the (Fe_2_Se_3_)_2_[Fe(dien)_2_]_0.9_ hybrid rods.

**Figure 4 nanomaterials-13-00487-f004:**
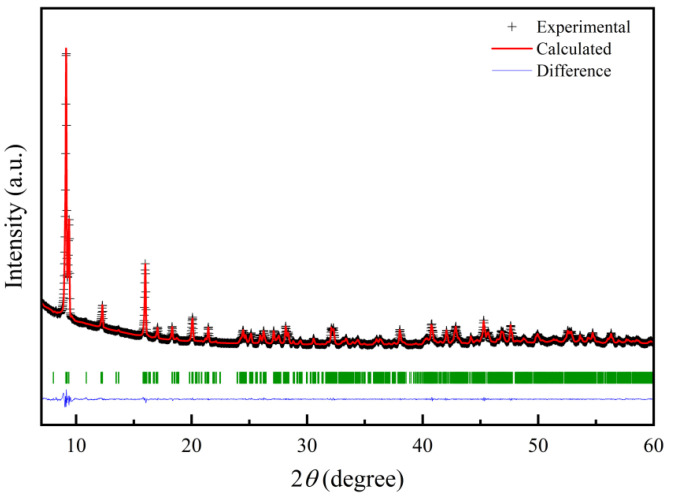
Powder XRD pattern of (Fe_2_Se_3_)_2_[Fe(dien)_2_]_0.9_ hybrid rods recorded at room temperature (solid crosses) with Rietveld refinements and difference curves.

**Figure 5 nanomaterials-13-00487-f005:**
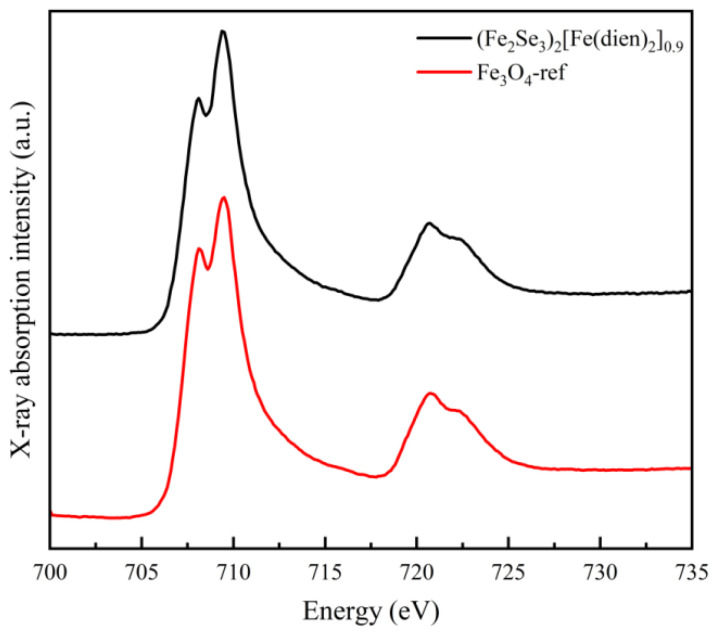
XAS of the (Fe_2_Se_3_)_2_[Fe(dien)_2_]_0.9_ hybrid rods and Fe_3_O_4_ as reference. The main peak of Fe *L*_3_-absorption edge is at 710 eV with lower shoulder at 708 eV, and the main peak of Fe *L*_2_-absorption edge is at 720 eV with higher shoulder at 722 eV.

**Figure 6 nanomaterials-13-00487-f006:**
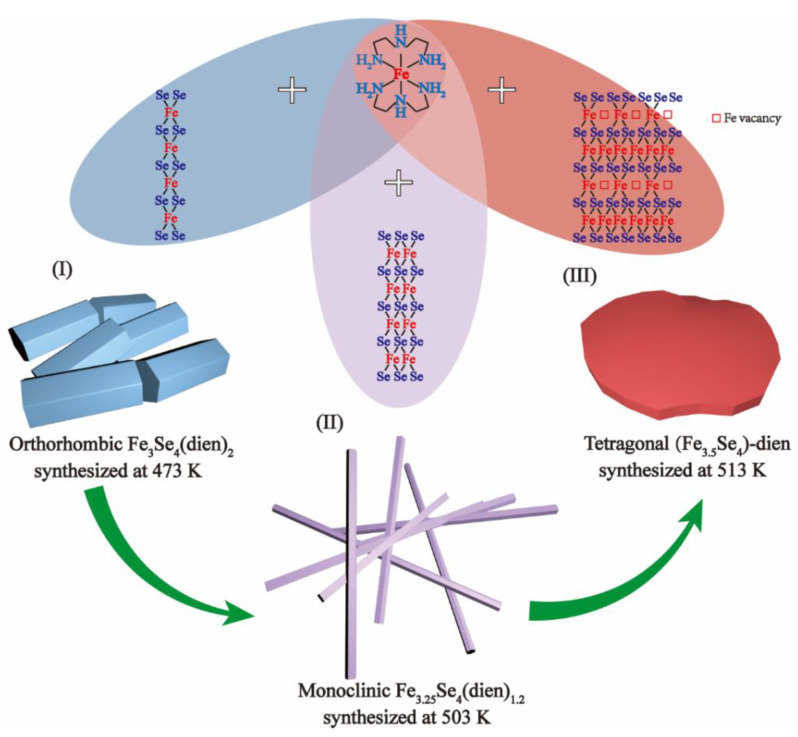
Schematic illustration of combining organic Fe(dien)_2_ complexes with (**I**) 1D FeSe_2_ chains, (**II**) Fe_2_Se_3_ double chains and (**III**) 2D Fe*_x_*Se layers, respectively, for the syntheses of three hybrid phases at different reaction conditions.

**Figure 7 nanomaterials-13-00487-f007:**
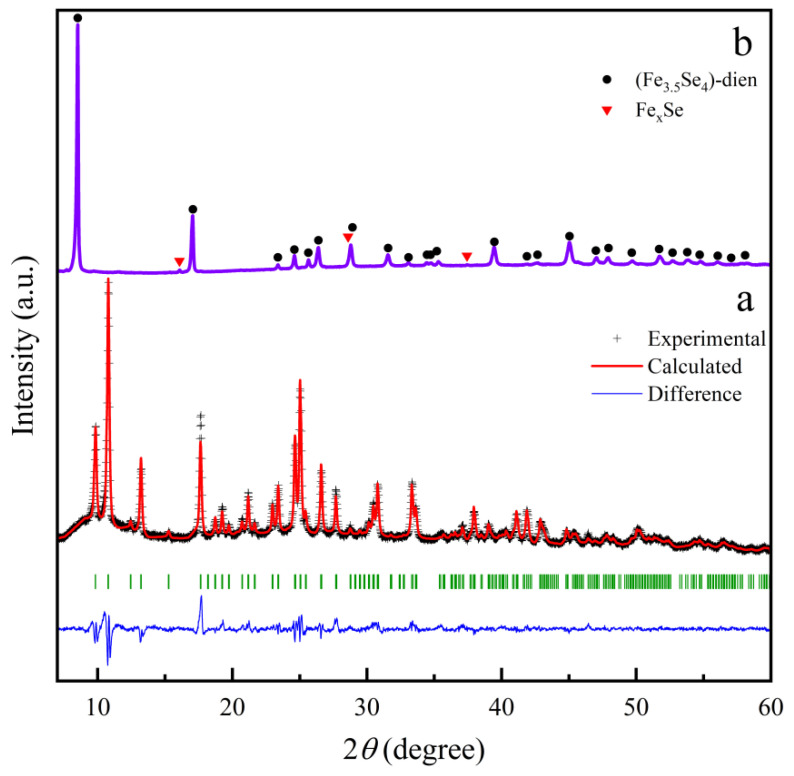
Powder XRD patterns of (**a**) [FeSe_2_]_2_[Fe(dien)_2_] particles and (**b**) (Fe_3.5_Se_4_)-dien plates.

**Figure 8 nanomaterials-13-00487-f008:**
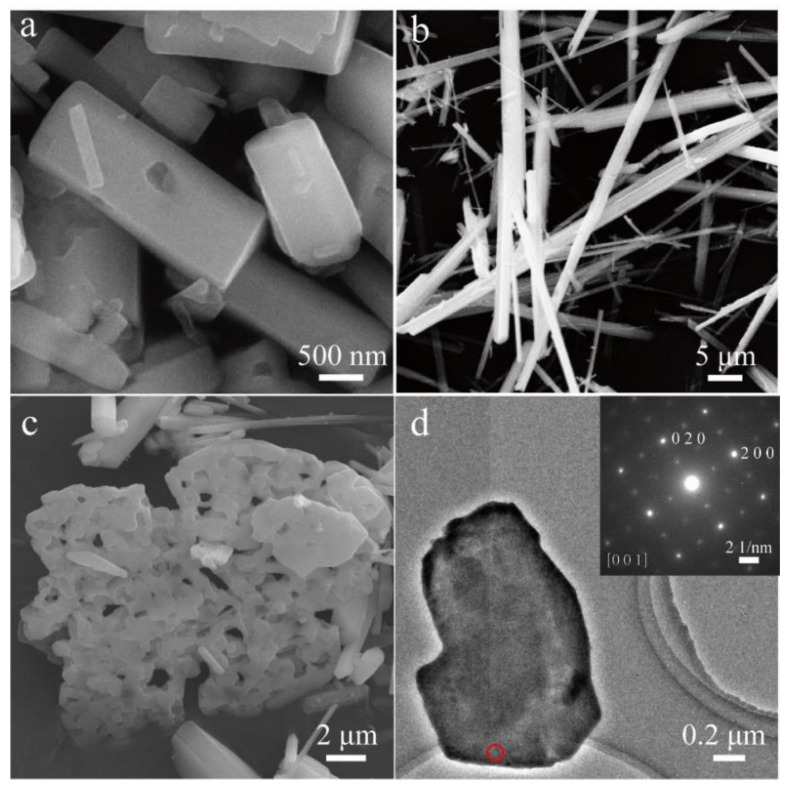
Microstructure of (**a**) [FeSe_2_]_2_[Fe(dien)_2_] hybrid particles, (**b**) (Fe_2_Se_3_)_2_[Fe(dien)_2_]_0.9_ hybrid rods, (**c**) an intermediate product between the rodlike and the platelike samples and (**d**) a (Fe_3.5_Se_4_)-dien hybrid plate. Inset shows the corresponding SAED pattern obtained perpendicular to the nanoplate marked by the red circle.

**Figure 9 nanomaterials-13-00487-f009:**
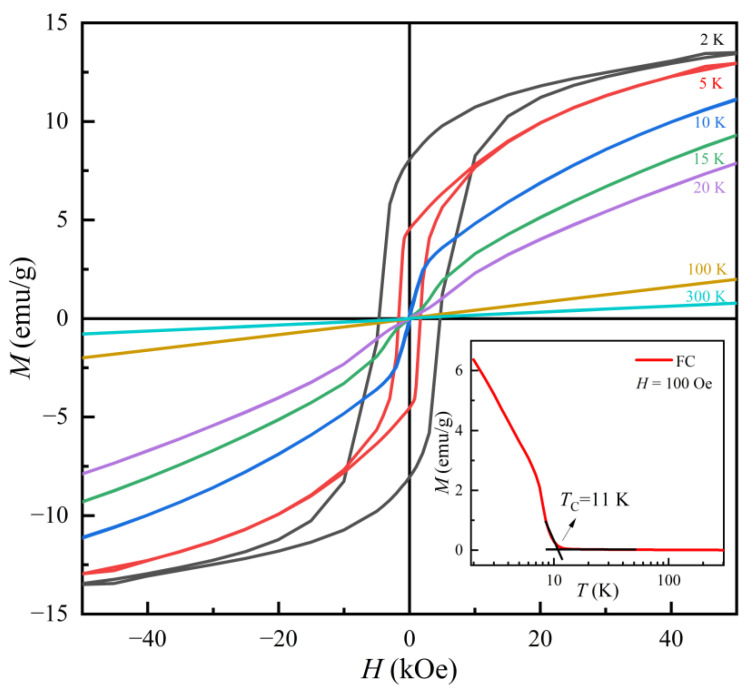
Hysteresis loops of (Fe_2_Se_3_)_2_[Fe(dien)_2_]_0.9_ hybrid rods measured at different temperatures. Inset shows temperature dependence of FC magnetizations in a magnetic field *H* = 100 Oe.

## Data Availability

The data are available upon reasonable request from the corresponding author.

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
