# Peer review of "Controllable Connection of Fe2Se3 Double Chains and Fe(dien)2 Complexes for Organic–Inorganic Hybrid Ferrimagnet with a Large Coercivity"

_nanomaterials, 2023, doi:10.3390/nano13030487_

Round 1
Reviewer 1 Report
In this paper, Zhang’s group developed a novel hybrid (Fe2Se3)2[Fe(dien)2]0.9 with a rod-like shape, which have a large coercivity and excellent magnetic performance. In my opinion, this work can be published in Nanomaterials after minor revision.
The comments for this paper are listed as below:
1. Please cite more references published in recent three years in the introduction part.
2. The authors should compare their results with the literature and cite references, such as synthesis method, magnetization and coercivity.
3. What applications is this hybrid material potentially applied in?
Reviewer 2 Report
The paper by Shang et al. is titled as “Controllable connection of Fe2Se3 double chains and Fe(dien)2 complexes for organic-inorganic hybrid ferrimagnet with a large coercivity”
It is understood, that the inorganic-organic hybrids can magnetize, but it is not that well understood, based on the results of the present manuscript, how can such structures possibly be fabricated in a controllable manner.
Round 2
Reviewer 2 Report
The manuscript has definitely been improved along with the review process.